# The Case of the Scribble Polarity Module in Asymmetric Neuroblast Division in Development and Tumorigenesis

**DOI:** 10.3390/ijms21082865

**Published:** 2020-04-20

**Authors:** Ana Carmena

**Affiliations:** Instituto de Neurociencias, Consejo Superior de Investigaciones Científicas/Universidad Miguel Hernández, 03550 Sant Joan d’Alacant, Alicante, Spain; acarmena@umh.es

**Keywords:** Scribble polarity module, asymmetric cell division, neuroblasts, tumorigenesis, *Drosophila*

## Abstract

The Scribble polarity module is composed by Scribble (Scrib), Discs large 1 (Dlg1) and Lethal (2) giant larvae (L(2)gl), a group of highly conserved neoplastic tumor suppressor genes (TSGs) from flies to humans. Even though the Scribble module has been profusely studied in epithelial cell polarity, the number of tissues and processes in which it is involved is increasingly growing. Here we discuss the role of the Scribble module in the asymmetric division of *Drosophila* neuroblasts (NBs), as well as the underlying mechanisms by which those TSGs act in this process. Finally, we also describe what we know about the consequences of mutating these genes in impairing the process of asymmetric NB division and promoting tumor-like overgrowth.

## 1. The Scribble Polarity Module

The Scribble polarity module is composed by Scribble (Scrib), Discs large 1 (Dlg1) and lethal (2) giant larvae (L(2)gl), all of which represent highly conserved neoplastic tumor suppressor genes (TSGs) from flies to humans [1,2,3,4]. *l(2)gl* alleles were first isolated in *Drosophila* by Bridges in 1930s [5], but their malignant mutant phenotype, an extensive overproliferation and tissue disorganization in imaginal epithelia and adult brain anlagen (optic lobe), was described years later in an spontaneous *l(2)gl* mutation (*l(2)gl^4^*) [6]. Thus, *l(2)gl* was the first example of a TSG, even before the term “tumor suppressor gene” came on stage [7,8,9], although the existence of a recessive class of cancer genes (“recessive oncogenes”) had already been suggested by Harris’ cell fusion experiments and Knudson’s retinoblastoma studies [10,11]. *dlg1* was isolated few years later in a screen for mutant phenotypes in the imaginal discs that altered their morphology. *dlg1* mutants showed hypertrophied and disorganized discs, a phenotype, as the authors pointed out, very similar to that shown by *l(2)gl^4^* mutants (Stewart et al. 1972). The third component of the module, *scrib*, was identified twenty years ago in a screen for mutations that altered the embryonic epithelial architecture in *Drosophila* [12]. *scrib* phenotype was identical to that shown by *l(2)gl* and *dlg1* mutants in embryonic, follicular and imaginal disc epithelia, strongly suggesting that all three genes acted in the same pathway to regulate cell growth and cell polarity [13]. A collaborative activity in a single pathway of *scrib*, *dlg1* and *l(2)gl* was confirmed by epistatic and strong genetic interactions between them, as well as by their cell localization and activity interdependence [13]. This, along with other previous works showed that the three neoplastic TSGs of the Scribble module were also key regulators of apico-basal cell polarity [3,12,13,14,15,16]. In epithelia, mutants in all these neoplastic TSGs lead to massive overgrowth and tumor formation in a homozygote condition, i.e., when the whole epithelial tissue is mutant for the TSG. However, mutant clones for each of these TSGs, *scrib, dlg1* and *l(2)gl*, do not overgrow, have poor survival, and are finally eliminated by cell competition, a process in which the surrounding wild-type cells activate a Jun N-terminal kinase (JNK)-mediated apoptosis in the mutant cells [17,18,19,20,21,22]. An oncogenic, activated form of Ras, *Ras^V1^*^2^, can prevent the cell death in these TSG single mutant clones by transforming the pro-apoptotic function of JNK into a pro-growth effect, inducing the formation of big neoplastic tumoral masses [18,19,21,23]. More recently, though, it has been shown, for four different *l(2)gl* null mutant alleles, that *l(2)gl* single mutant clones in eye-antennal discs behave differently from *scrib/dlg1* clones, in the sense that they are not eliminated by cell competition. In fact, *l(2)gl* clones overproliferate without affecting cell polarity by repressing the Hippo signaling pathway [24,25]. Intriguingly, it has been found that not only *l(2)gl* but also *dlg1* single mutant clones, in antennal discs, overgrow by activating the Ras-MAPK signaling pathway, while *scrib* clones do not [26]. Thus, it might be that the consequences of losing these TSGs are context-dependent, and that the induced levels of signaling pathways, such as JNK or Ras-MAPK, in the mutant clones determine the apoptosis or survival of the clone.

Scrib, Dlg1, and L(2)gl are all cytoplasmic, membrane-associated proteins that co-localize to the baso-lateral domain of epithelial cells. Scrib and Dlg1 are multi PDZ (PSD-95/Dlg1/ZO-1) domain-containing proteins that act as scaffolds, whereas L(2)gl is a non-muscle myosin II binding protein with WD-40 repeats and, unlike Scrib and Dlg1, is cortically uniform (Figure 1) [1,3,12,22,27,28].

Some physical interactions between the Scribble module proteins have been described in mammals [29,30], and in *Drosophila* the adaptor protein GUK-holder (Gukh) links Scrib and Dlg1 [31]. In addition, multiple proteins directly interact to each of the core components of the Scribble module in many different cellular contexts and processes, and the number of interacting proteins is increasingly growing [4]. Among these proteins are included regulators of different types of cell polarity, cytoskeleton, vesicular trafficking, tissue growth, as well as cell proliferation, survival, and migration [3,4]. Some of these proteins might be potential novel regulators of asymmetric neuroblast (NB) division, another process in which the Scribble module is also operative.

## 2. Beyond Epithelia: The Scribble Polarity Module in Neuroblasts

### 2.1. Asymmetric Division of Drosophila Neuroblasts

NBs, the neural stem cells of the *Drosophila* central nervous system (CNS), divide asymmetrically to give rise to another NB that keeps on dividing and a daughter cell called ganglion mother cell (GMC) that will start a differentiation program [32,33,34]. This cell fate commitment is possible by the action of cell-fate determinants, which are asymmetrically located at the basal pole of metaphase NBs and segregate exclusively to the GMC during NB division (Figure 2). The translational regulator brain tumor (Brat), the transcription factor Prospero (Pros), and the cytoplasmic protein Numb are among those determinants that inhibit proliferation and activate differentiation in the GMC [35,36,37,38,39,40,41,42,43].

A group of proteins located at the apical cortex of metaphase NBs control, in turn, the basal sorting of cell-fate determinants, as well as the orientation of the mitotic spindle along the NB apico-basal axis of polarity, two key processes to ensure the asymmetry of the division. This apical complex is an intricate protein network that includes the conserved partitioning defective proteins Par-6 and Par-3 (Bazooka, Baz, in *Drosophila*) and the atypical protein kinase C (aPKC) (Figure 2) [44,45,46,47,48]. Baz physically interacts with the adaptor protein Inscuteable (Insc) that in turn binds and activates Partner of Insc (Pins; LGN in mammals), allowing the interaction between the Gαi protein subunit anchored to the membrane and Pins, which thereafter orchestrates the orientation of the spindle (Figure 2) [49,50,51,52,53,54,55,56]. This process requires the function of Canoe (Cno; Afadin in mammals) that, after being phosphorylated by the serine-threonine kinase Warts (Wts; LATS1-2 in mammals), binds the N-terminal Pins^TPR^ domain, the same region that Insc was bound to [56,57,58,59]. Cno then contributes to the apical recruitment of the Pins-interacting proteins Mushroon body defect (Mud; NuMA in mammals) and Dlg1 [57,58]. Dlg1 binds the middle Pins^LINKER^ domain and the Kinesin heavy chain 73 (Khc-73) motor protein that interacts with astral microtubule plus-ends, anchoring the spindle to the apical cortex [60,61,62]. Mud, like Cno, interacts with the Pins^TPR^ domain and, additionally, with the Dynein molecular motor, which binds the astral microtubule minus-ends promoting pulling forces on them and reinforcing the apical-basal orientation of the spindle [60] (Figure 2).

### 2.2. Types of Neuroblasts: Different Lineages, Same Origin

Embryonic NBs delaminate from the neuroectoderm inheriting the apico-basal polarity of the neuroepithelial cells. The establishment of an axis of cell polarity is a prerequisite for a correct asymmetric division. Once this axis of cell polarity is established, the mitotic spindle aligns along it and the cell-fate determinants localize asymmetrically at the basal pole of the NB. These embryonic NBs will divide a finite number of times, up to twenty, entering quiescence at the end of embryogenesis. At late first larval stage, NBs resume proliferation, this time undergoing hundreds of them and increasing their size before each division. These NBs that divide to give rise to another NB and a GMC have been called type I NBs (Figure 3) [33]. Some years ago, another type of NBs, called type II NBs, were found in the larval central brain [37,63,64]. These NBs also divide asymmetrically to give rise to another NB and, instead of a GMC, a progenitor cell called an intermediate progenitor (INP) that, after a maturation process, will divide asymmetrically to give rise to another INP and a GMC (Figure 3). Given this additional phase of proliferation, type II NB lineages are larger than type I and more prone to induce tumor-like overgrowth when the process of ACD is compromised (see below, Section 2.4). In addition, while type I NB lineages occupy most of the central brain, these type II NB lineages are only eight per brain hemisphere and are located at precise locations at the dorso-medial part of the brain (Figure 3). Very recently, it has been shown that type II NBs have also an embryonic origin and are arrested at the end of embryogenesis [65,66].

### 2.3. The Scribble Module in Asymmetric Neuroblast Division during Development

A role for the neoplastic TSGs of the Scribble module in asymmetric NB division was first shown for Dlg1 and L(2)gl [67,68]. Dlg1 and L(2)gl were found to be essential for the basal targeting of the cell-fate determinants Numb and Pros, as well as of their adaptor proteins Partner of Numb (Pon) and Miranda (Mira), respectively, in both embryonic and larval mitotic NBs [67,68]. However, Dlg1 and L(2)gl were dispensable for the localization of apical proteins, such as Baz, Insc or Pins and for the orientation of the mitotic spindle [67,68]. Dlg1 was required for the cortical localization of L(2)gl, which became cytoplasmic in *dlg1* mutant embryos; however, L(2)gl was not necessary for the localization of Dlg1. Hence, it was proposed that, at least for its localization, although not necessarily for its function, Dlg1 would act upstream of L(2)gl [67,68]. In fact, we now know that L(2)gl acts functionally upstream of Dlg1 [69]. Both proteins are distributed predominantly at the cortex, although, at metaphase, Dlg1 is apically enriched while L(2)gl is phosphorylated and inactivated by aPKC at this location [70]. This is promoted by Aurora-A (AurA) kinase, which at metaphase phosphorylates Par-6 with the consequent activation of aPKC. Activated aPKC phosphorylates and inactivates L(2)gl, which leaves the apical complex and it is replaced by Baz/Par-3 [69,71]. Baz, then, allows the phosphorylation of the cell-fate determinant Numb by aPKC, and the consequent exclusion of P-Numb to the basal pole of the NB [69] (Figure 4a). The inhibition of L(2)gl by aPKC is mutual, as L(2)gl represses aPKC basally, restricting it to the apical cortex [72] (Figure 4b,c). Thus, the localization of at least some apical proteins, such as aPKC, do depend on some of the TSGs of the Scribble module. L(2)gl also binds and represses non-muscle myosin II heavy chain, called Zipper in *Drosophila*, at interphase. At metaphase, when L(2)gl is inactivated by aPKC, it was proposed that myosin II becomes active and, in turn, promotes the cortical exclusion of the cell-fate determinant adaptor protein Mira from the apical NB cortex (Figure 4b) [28,73]. The basal targeting of Mira would occur by passive diffusion throughout the cytoplasm, not by active transport, and it would depend on another myosin, myosin VI, Jaguar in *Drosophila*, which would be essential for the final localization of Mira in a basal crescent (Figure 4b) [74,75]. Yet, the role of myosin II in Mira localization (Figure 4b) was questioned and the model to explain Mira asymmetry was replaced by another one some years ago [76]. This latter work showed that aPKC can directly phosphorylate Mira at several sites to exclude it from the apical cortex independently of L(2)gl, which would be antagonizing aPKC activity (Figure 4c) [71,76]. More recently, additional data seem to point to an integrated view of both models [77]. Thus, aPKC direct phosphorylation of Mira, event that occurs at prophase, would not be the only mechanism that regulates Mira asymmetry, and an actomyosin-dependent mechanism would be additionally required to maintain Mira asymmetric localization at metaphase (Figure 4d) [77,78].

Regarding Dlg1, over the past 20 years, since it was first described in the process of NB asymmetric division [67,68], we have substantially increased our knowledge about the mechanisms underlying Dlg1 function in this context. The guanylate kinase (GK) domain of Dlg1/DLG1 (Figure 5a), a phosphoprotein recognition motif, binds the Pins/LGN linker domain (Pins^Linker^) both in *Drosophila* and in mammals [60,79,80]. This conserved Pins^Linker^ domain must be phosphorylated by the mitotic kinase AurA to physically interact with the Dlg1 GK domain [60], which in turn recruits the motor protein Khc-73. This kinase first interacts through its MAGUK binding stalk (MBS) domain (Khc-73^MBS^) with the GK motif of Dlg1 at the cortex, and then with astral microtubule plus-ends through its motor domain (Khc-73^motor^) (Figure 5a). This Pins-Dlg1-Khc73 pathway mediates a microtubule-induced Pins-Gαi (the latter is bound to the GoLoco domains of Pins, see above and Figure 2) cortical polarity at metaphase NBs, independently of the Par complex [61]. However, this pathway is not enough for a full orientation of the mitotic spindle. Pins must activate another microtubule motor pathway mediated by Dynein that interacts with minus-end astral microtubules. The Pins^TPR^ domain is the motif involved in the activation of this pathway by binding Mud/NuMA, which in turn associates with Dynein that exerts pulling forces on microtubules. Both Pins^TPR^- and Pins^Linker^-mediated pathways are required and act synergistically to promote a robust spindle alignment [60]. The mechanism by which these Pins-mediated pathways interact was identified some years ago [81] (Figure 5b). In this work, authors show how the *Drosophila* 14-3-3ζ protein associates to the 14-3-3 binding motif present in the Khc-73 C-terminal stalk, (Figure 5a). The NudE Dynein cofactor [82,83] interacts in turn with 14-3-3ε, which forms a heterodimer with 14-3-3ζ. This complex 14-3-3ζ/14-3-3ε/NudE acts then as the bridge between both Pins-mediated pathways to allow a full, optimal spindle orientation (Figure 5b) [81]. More recently, Dlg1 has been shown to be phosphorylated in its SH3 domain by aPKC [84] (Figure 5c). This phosphorylation releases an auto-inhibitory intramolecular interaction between Dlg1 SH3 and the GUK domains. In this situation, the spindle orientation factor Gukh can bind to the Dlg1 GUK domain and to astral microtubules, contributing, along other Dlg1 effectors such as Khc-73, to Dlg1-mediated spindle alignment (Figure 5c).

As mentioned above, Scrib was identified later than L(2)gl and Dlg1 [13] and consequently, it was described to be involved in NB asymmetric division a posteriori than those ACD regulators [85]. In this work, Scrib localization was found to be cortical in NBs, with an apical enrichment at metaphase, similar to Dlg1 distribution. Likewise, as L(2)gl, Scrib localization was dependent on Dlg1 [85]. Authors described for the first time the function of all these TSGs, L(2)gl, Dlg1 and Scrib, in regulating cell size and mitotic spindle asymmetry in NBs. While in wild-type telophase NBs, the NB was bigger than the GMC, and the apical centrosome and astral microtubules larger than the basal ones, in *l(2)gl*, *dlg1* and *scrib* embryonic mutant NBs, symmetric divisions (with equal-sized NB and GMC) and even inverted divisions (with the NB smaller than the GMC) were detected [85]. Scrib, as previously shown for L(2)gl and Dlg1, was found to be required for basal targeting of cell fate determinants and adaptor proteins, such as Mira and Pros, but not for the localization of apical proteins [85]. More recently, however, the apical protein aPKC has been shown to require Scrib for a proper cortical crescent formation at metaphase in type II NB lineages of the larval brain [26]. Thus, over the past years, all of these TSGs (L(2)gl, Dlg1, and Scrib) have been shown to be also necessary for the correct localization of at least some apical proteins (i.e., L(2)gl for aPKC; Dlg1 for Pins and Scrib for aPKC localization). Some of the Scrib motifs, such as the LRR region and the PDZ domains (Figure 1) have been proved to be required for the proper cortical localization and function of Scrib in NBs [86]. However, while the mechanisms by which L(2)gl and Dlg1 regulate the asymmetric division of NBs have been deeply investigated over the past years, we do not have any clear clue about the underlying mechanisms or mode of action of Scrib in this context.

### 2.4. The Scribble Module in Asymmetric Neuroblast Division during Tumorigenesis

ACD is a fundamental process during development to generate cell diversity. In addition, as we have learned over the past years, ACD is also a relevant process to take into account in the context of cancer and stem cell biology. A connection between failures in the process of ACD and tumorigenesis was first shown in the lab of C. González using the neural stem cells or NBs of the *Drosophila* larval brain as a model system [87]. In these experiments, pieces of GFP-labeled brains mutant for different ACD regulators were transplanted into the abdomen of adult host flies. These flies, after several weeks, developed big tumoral masses inside their abdomen, tumors that in some cases induced metastatic growth [87]. However, mutations in genes involved in ACD modulation do not always cause tumor-like overgrowth. It will depend on the type of ACD regulator and the particular environment in which the NB lineage grows [88]. For example, type II NB clones in the larval brain mutant for the ACD regulator gene *cno*/*AFDN* or for each of the Scribble module (*l(2)gl*, *dlg1* and *scrib*) do show ectopic NBs within the clone but they do not overgrow [26]. In fact, at least the *scrib* mutant clones are smaller than control NB clones and they do not appear very frequently. As it occurs in epithelial *scrib* mutant clones, in *scrib* NB clones a JNK activity-dependent apoptosis is also triggered [26]. However, the simultaneous loss of *scrib* and *cno/AFDN* in these larval NB clones overcomes the *scrib*/JNK-induced apoptosis and causes massive tumor-like overgrowths [26]. This effect is due to the upregulation of Ras, normally repressed by Cno/Afadin [89,90]. Activated Ras, then, promotes a switch in the JNK function, from a pro-apoptotic to a pro-growth effect, similar to what occurs in epithelial *Ras^V12^ scrib* double mutant clones [18,19,23,26]. Neither *cno l(2)gl* nor *cno dlg1* double mutant clones show the strong synergistic cooperation displayed in *cno scrib* mutant clones. In fact, the former double mutant clones are very similar to *cno* single mutant clones [26]. One possibility to explain the different behavior of *cno l(2)gl* and *cno dlg1* mutant clones is that JNK is not activated in *l(2)gl* nor in *dlg1* NB single mutant clones, even though in epithelia JNK is activated in each of those single mutant clones [19]. This is something that should be analyzed in detail in NB mutant clones. Nevertheless, the capability or not of inducing JNK in the *l(2)gl* or *dlg1* single NB mutant clones is probably not the only explanation, as *Ras^V12^ scrib* NB mutant clones do not show the tumor-like overgrowth shown by *cno scrib* NB mutant clones [26]. Thus, altogether, the data we currently have strongly suggest that Cno is acting in the same pathway that the ACD regulators Dlg1 and L(2)gl and is epistatic to them. This is consistent with previous results showing that Cno contributes to Dlg1 recruitment to the apical pole of the NB [57] and that Cno is required for a proper aPKC cortical localization [26], as aPKC acts upstream of L(2)gl [69]. However, Scrib must be working in at least a partially independent pathway to that involving the ACD regulators, Cno, L(2)gl, and Dlg1, and this would explain the strongest effect of *cno scrib* double mutant clones. Hence, in asymmetric NB division, the Scribble module does not seem to be so functionally interdependent as in epithelia.

## 3. Conclusions

The Scribble module has been thoroughly analyzed in epithelial cell polarity. However, over the past decades this group of TSGs has been involved in additional processes and in different contexts, including the asymmetric division of *Drosophila* neural stem cells. In this process, L(2)gl regulates cell-fate determinant localization by counteracting aPKC activity, and Dlg1 is a key component of the spindle orientation machinery by interacting with different microtubule-binding proteins, such as Khc-73 and Gukh. Nevertheless, while the mechanisms by which L(2)gl and Dlg1 modulate asymmetric NB division have been elucidated in more detail, the mode of Scrib action remains to be unveiled. Future work will be necessary to enlighten this point, as well as to further clarify the interdependence of these TSGs during asymmetric NB division.

## Figures and Tables

**Figure 1 ijms-21-02865-f001:**
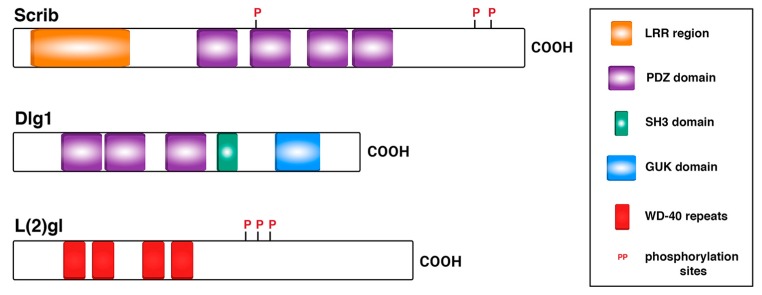
The Scribble polarity module. The modular structure of the proteins belonging to the Scribble module, Scrib, Dlg1, and L(2)gl, is shown. The LAP (Leucine-rich repeats Aand PDZ domains) protein Scrib and the MAGUK (Membrane Associated GUanylate Kinase) protein Dlg1 both contain PDZ domains and additional motifs, such as 16 Leucine-Rich-Repeats (LRRs, in Scrib) or a Src Homology 3 and a GUuanylate Kinase (SH3 and GUK domains, in Dlg1). L(2)gl contains WD-40 repeats. Conserved phosphorylation sites are shown in both L(2)gl and Scrib.

**Figure 2 ijms-21-02865-f002:**
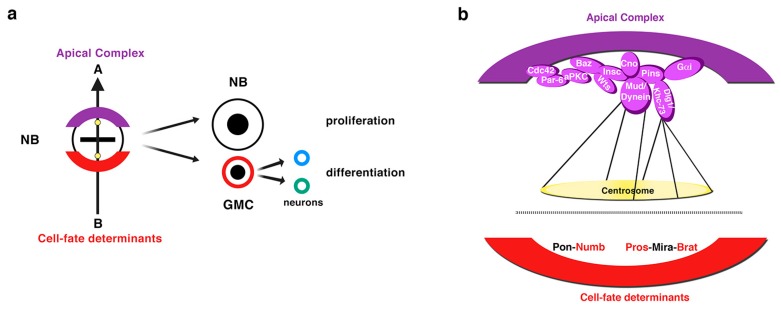
*Drosophila* neuroblasts (NBs), the neural stem cells of the central nervous system (CNS), divide asymmetrically. (**a**) NBs divide asymmetrically to give rise to another NB and a ganglion mother cell (GMC), which receives the cell-fate determinants that induce a differentiation program in this cell. The GMC divides asymmetrically through a terminal division to give rise to two different neurons of glial cells. The sibling NB that does not receive the cell-fate determinants keeps on dividing. A group of proteins apically located at the cortex of metaphase NBs (the “apical complex”) is in turn crucial for the basal sorting of the cell-fate determinants, as well as for the correct orientation of the mitotic spindle along an apico-basal axis of cell polarity previously established. (**b**) A diagram showing the most representative components of the apical complex and the cell-fate determinants Numb, Pros and Brat. Pon and Mira are adaptor proteins of Numb (Pon) and of Pros and Brat (Mira) (modified from Carmena, Fly, 2018).

**Figure 3 ijms-21-02865-f003:**
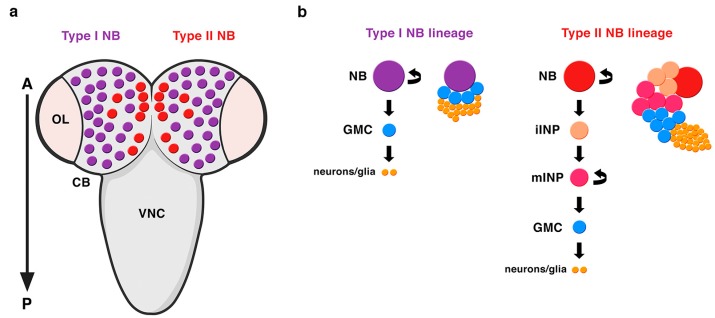
Types of NBs in the *Drosophila* CNS. (**a**) A dorsal view of the larval central brain (CB) containing type I (purple) and type II (red) NBs. There are only eight type II NB lineages per brain hemisphere located at very specific positions at the dorso-medial part of the CB. OL: optic lobe; VNC: ventral nerve cord; A: anterior; P: posterior. (**b**) Type II NB lineages are bigger than type I NB lineages. In type II NB lineages, the NB divides asymmetrically to generate another NB and, instead of a GMC (like in type I NB lineages), an intermediate progenitor (INP), which after a process of maturation, divides asymmetrically to give rise to another INP and a GMC. iINP: immature INP; mINP: mature INP (modified from Carmena, Fly, 2018).

**Figure 4 ijms-21-02865-f004:**
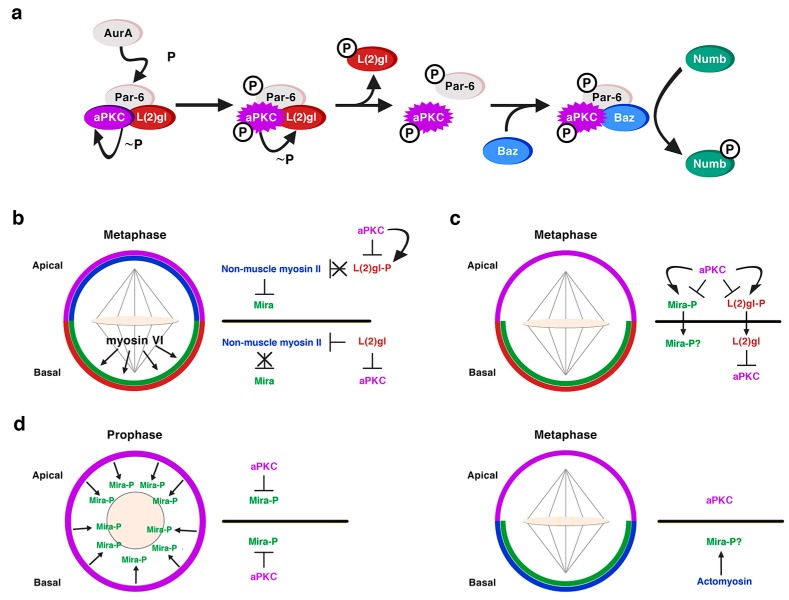
L(2)gl in asymmetric NB division. (**a**) L(2)gl forms part of an inactive Par complex. At metaphase, the kinase AurA phosphorylates Par6, which leads to the activation of aPKC and the consequent phosphorylation of L(2)gl by active aPKC. P-L(2)gl then leaves the Par complex and it is replaced by Baz/Par-3, which binds both aPKC and Numb making possible the phosphorylation of Numb by aPKC and the exclusion of P-Numb from the apical cortex. (Modified from Wirtz-Peitz et al., Cell, 2008). (**b**) Myosin-dependent model to explain the basal sorting of the adaptor protein Mira. aPKC phosphorylates and inactivates L(2)gl at the apical pole of metaphase NBs. Hence L(2)gl cannot bind and inactivate myosin II, which excludes Mira from the apical cortex. Myosin VI would help to locate Mira in a basal crescent. L(2)gl is active at the basal pole inhibiting both aPKC and myosin II, allowing in this way the accumulation of Mira at this location. (**c**) Myosin-independent model to explain the basal sorting of Mira. Apical aPKC directly phosphorylates both L(2)gl and Mira excluding them from the apical cortex. At the basal pole L(2)gl counteracts the activity of aPKC. (**d**) An integrative model both aPKC and myosin-dependent. At prophase, before the nuclear membrane is disorganized, cortical aPKC phosphorylates Mira and excludes it from the cortex. At metaphase, aPKC is apically enriched and the basal actomyosin network contributes to the asymmetric Mira retention by providing an anchoring scaffold to Mira at this location. The role of L(2)gl is not discussed in the context of this model (Hannaford et al. eLife, 2018), but it could be also counteracting the activity of aPKC basally.

**Figure 5 ijms-21-02865-f005:**
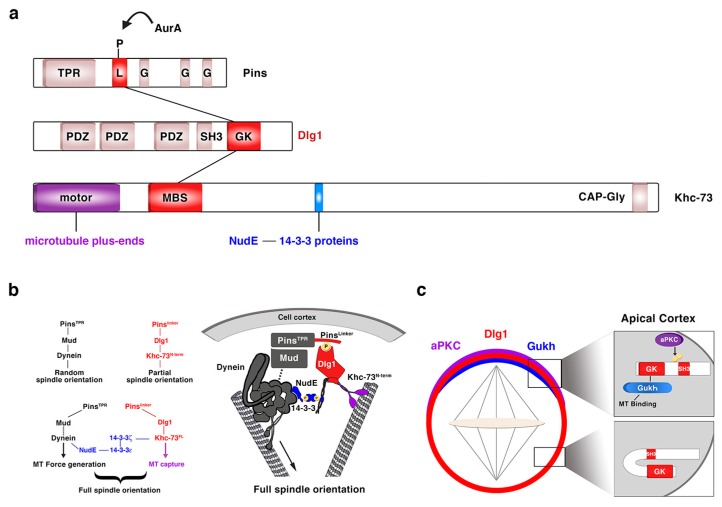
Dlg1 in asymmetric NB division. (**a**) Modular structure of the ACD regulators Pins, Dlg1 and Khc-73. The kinase AurA phosphorylates the linker domain (L) of Pins, and the GK domain of Dlg1 binds both this phosphorylated Pins^Linker^ domain and the MBS motif of Khc-73. This kinase binds astral microtubule plus-ends through its motor domain and 14-3-3ζ protein through a 14-3-3 binding motif present at the C-terminal stalk, between the MBS and the CAP-Gly motif. TPR: TetratricoPeptide Repeat; L: Linker; G: GoLoco; PDZ: PSD-95/Dlg/ZO-1; SH3: Src Homology 3; GK: Guanylate Kinase; MBS: Maguk Binding Stalk; CAP-Gly: Cytoskeleton Associated Proteins-Glycine-rich. (**b**) The two Pins-mediated pathways that orientate the mitotic spindle are connected through a NudE-14-3-3 protein bridge, which binds the two motor proteins involved in each of those pathways. NudE binds the motor Dynein and 14-3-3ε, which forms a heterodimer with 14-3-3ζ that in turn interacts with the motor Khc-73 (adapted from Lu and Prehoda, Dev Cell, 2013). (**c**) aPKC phosphorylates the SH3 domain of Dlg1 releasing an intramolecular inhibitory binding between SH3 and GK domains. GK can then bind the microtubule interactor protein Gukh, which contributes to the proper orientation of the mitotic spindle (adapted from Golub et al., eLIFE, 2017).

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
