# Peer review of "The Case of the Scribble Polarity Module in Asymmetric Neuroblast Division in Development and Tumorigenesis"

_ijms, 2020, doi:10.3390/ijms21082865_

Round 1

Reviewer 1 Report

This is well written review summarising concisely what is known about the Scribble module in asymmetric cell division in Drosophila neuroblast, a developmental system that has been key to understand the fundamental regulation of asymmetric cell divisions in vivo. The system is introduced and treated simply enough but in enough detail for non-specialist to be able to follow the information that follows. The review of the literature is thorough and highlights some of the contrasting models that have been proposed in the field with a view to resolve some of these disparities. These points are suitably illustrated throughout the figures which are all clear and add greatly to the review. I thoroughly enjoyed reading the review, this is a great addition to the field.

A few minor typos:

Line 31: “first example of TSG” should be “first example of aTSG”

Page 33: “by Harris cell fusion experiments” should eb “by Harris’ cell fusion experiments”

Page 34: “by Knudson retinoblastoma studies” should eb “by Knudson’s retinoblastoma studies”

Line 266: “as previously shown by L(2)gl and Dlg1” should be “as previously shown for L(2)gl and Dlg1”

Line 303: “that the JNK is not activated” should be “that JNK is not activated”

Author Response

Reviewer 1

A few minor typos:

Line 31: “first example of TSG” should be “first example of aTSG”

It has been corrected.

Line 33: “by Harris cell fusion experiments” should eb “by Harris’ cell fusion experiments”

It has been corrected.

Line 34: “by Knudson retinoblastoma studies” should eb “by Knudson’s retinoblastoma studies”

It has been corrected.

Line 293: “as previously shown by L(2)gl and Dlg1” should be “as previously shown for L(2)gl and Dlg1”
It has been corrected.

Line 330: “that the JNK is not activated” should be “that JNK is not activated”

It has been corrected.

Reviewer 2 Report

In this review, Carmena described the role of the Scribble polarity module in the asymmetric division of Drosophila neuroblasts. While the Scribble polarity module was a very important structure indicated to be involved in many processes, a review with a focus on its function on the asymmetric division of Drosophila neuroblasts is helpful. I have the following suggestions:

  1. For the structure of the review, I think it would be better to start with a more general introduction of the Scribble polarity module that would be interesting to the general audience, such as the indicated roles of the Scribble polarity module as “tumor suppressor gene” and their time of discovery and brief introduction of its indicated role in different cell types. After the general introduction, I feel it makes more sense to follow this section with what is described in section 2: the different structural domains of the scrib, Dlg1 and L2gl. After that, it is good to discuss the role of the scribble polarity module in epithelia cells, and then the role of scribble polarity module in neuroblast.

  1. Line 53: it would be helpful to explain what the l(2)gl single mutant clone is, so that reader doesn’t need to look into the reference to see whether it is L(2)gl truncation, knockout, or mutations in what sites.

  1. Line 85: It would be helpful to include proteins interacting with the scribble module protein that might be important for the asymmetric division, so that the connection to the next sentence in line 86 would be natural.

  1. In Section 2.2 types of neuroblasts, it would be helpful to mention the differences of the type1 NB and type II NB and its biological implications. Otherwise, this section reads a bit dry.

  1. Line 277: Could the author comment on the possible reasons for why and What kind of research will lead to understanding of Scribble’s mode of action?

  1. Line 316: In the conclusion section, it would be helpful to include a more detailed summary of mutant studies of Scribble module in different processes and its indication of its role, and why the mechanism is not conclusive and what further studies need to be done for this field would be helpful.

Author Response

Reviewer 2

1. For the structure of the review, I think it would be better to start with a more general introduction of the Scribble polarity module that would be interesting to the general audience, such as the indicated roles of the Scribble polarity module as “tumor suppressor gene” and their time of discovery and brief introduction of its indicated role in different cell types. After the general introduction, I feel it makes more sense to follow this section with what is described in section 2: the different structural domains of the scrib, Dlg1 and L2gl. After that, it is good to discuss the role of the scribble polarity module in epithelia cells, and then the role of scribble polarity module in neuroblast.

I have moved the description of the different structural domains of Scrib, Dlg1 and L(2)gl from section 2 to section 1, as the reviewer suggested

2. Line 53: it would be helpful to explain what the l(2)gl single mutant clone is, so that reader doesn’t need to look into the reference to see whether it is L(2)gl truncation, knockout, or mutations in what sites.

I have added a brief explanation about the nature of the l(2)glmutant alleles used. It appears in line 53: “(…) shown, for four different l(2)gl null mutant alleles, that…”.

3. Line 85: It would be helpful to include proteins interacting with the scribble module protein that might be important for the asymmetric division, so that the connection to the next sentence in line 86 would be natural.

I have included an alternative sentence in line 86: “Among these proteins are included regulators of different types of cell polarity, cytoskeleton, vesicular trafficking, tissue growth, as well as cell proliferation, survival and migration [3,4]. Some of these proteins might be potential novel regulators of asymmetric neuroblast (NB) division, another process in which the Scribble module is also operative.”

4. In Section 2.2 types of neuroblasts, it would be helpful to mention the differences of the type1 NB and type II NB and its biological implications. Otherwise, this section reads a bit dry.

I have added some additional information in lines 153-156: “These NBs that divide to give rise to another NB and a GMC have been called type I NBs (Figure 3) [33]. Some years ago, another type of NBs, called type II NBs, were found in the larval central brain [37,63,64]. These NBs also divide asymmetrically to give rise to another NB and, instead of a GMC, a progenitor cell called an intermediate progenitor (INP) that, after a maturation process, will divide asymmetrically to give rise to another INP and a GMC (Figure 3). Given this additional phase of proliferation, type II NB lineages are larger than type I and more prone to induce tumor-like overgrowth when the process of ACD is compromised (see later, section 2.4). In addition, while type I NB lineages occupy most of the central brain, these type II NB lineages, are only eight per brain hemisphere, and are located at precise locations at the dorso-medial part of the brain (Figure 3).

5. Line 277: Could the author comment on the possible reasons for why and What kind of research will lead to understanding of Scribble’s mode of action?

Our recent work on the role of Scrib regulating type II NB asymmetric division in cooperation with other ACD regulators, made us realizing that we do not know basically nothing about the mechanism by which Scrib functions in this context. Given the relevance of the Scrib module in other tissues, we think this is a relevant issue to address, so we are trying to investigate this in the lab using different approaches.

6. Line 316: In the conclusion section, it would be helpful to include a more detailed summary of mutant studies of Scribble module in different processes and its indication of its role, and why the mechanism is not conclusive and what further studies need to be done for this field would be helpful.

I have added in line 349, “Conclusions section”, additional sentences summarizing what we know about the role of L(2)gl and Dlg1 in NB asymmetric division.